# Dynamic Interactions between Diarrhoeagenic Enteroaggregative *Escherichia coli* and Presumptive Probiotic Bacteria: Implications for Gastrointestinal Health

**DOI:** 10.3390/microorganisms11122942

**Published:** 2023-12-08

**Authors:** Wisdom Selorm Kofi Agbemavor, Elna Maria Buys

**Affiliations:** 1Department of Consumer and Food Sciences, University of Pretoria, Private Bag X20, Pretoria 0028, South Africa; 2Radiation Technology Centre, Biotechnology and Nuclear Agriculture Research Institute, Ghana Atomic Energy Commission, Legon, Accra P.O. Box LG 80, Ghana

**Keywords:** EAEC strains, LAB strains, inflammation, intestinal barrier, tight junctions, cytokine secretion, interleukin-8 (IL-8), bacterial infection dose (BID), epithelial cells, adhesion behaviour

## Abstract

This study delves into the temporal dynamics of bacterial interactions in the gastrointestinal tract, focusing on how probiotic strains and pathogenic bacteria influence each other and human health. This research explores adhesion, competitive exclusion, displacement, and inhibition of selected diarrhoeagenic *Escherichia coli* (D-EAEC) and potential probiotic strains under various conditions. Key findings reveal that adhesion is time-dependent, with both D-EAEC K2 and probiotic *L. plantarum* FS2 showing increased adhesion over time. Surprisingly, *L. plantarum* FS2 outperformed D-EAEC K2 in adhesion and exhibited competitive exclusion and displacement, with inhibition of adhesion surpassing competitive exclusion. This highlights probiotics’ potential to slow pathogen attachment when not in competition. Pre-infecting with *L. plantarum* FS2 before pathogenic infection effectively inhibited adhesion, indicating probiotics’ ability to prevent pathogen attachment. Additionally, adhesion correlated strongly with interleukin-8 (IL-8) secretion, linking it to the host’s inflammatory response. Conversely, IL-8 secretion negatively correlated with trans-epithelial electrical resistance (TEER), suggesting a connection between tight junction disruption and increased inflammation. These insights offer valuable knowledge about the temporal dynamics of gut bacteria interactions and highlight probiotics’ potential in competitive exclusion and inhibiting pathogenic bacteria, contributing to strategies for maintaining gastrointestinal health and preventing infections.

## 1. Introduction

The intricate interplay between microbial communities within the human gastrointestinal tract has emerged as a pivotal factor influencing overall health [1,2]. Among the diverse array of microbes inhabiting the gut, enteroaggregative *Escherichia coli* (EAEC) has drawn particular attention due to its pathogenic nature and its association with gastrointestinal infections. EAEC stands out as a significant contributor to diarrheal diseases worldwide [3,4]. Recognized for its ability to adhere to the intestinal mucosa, EAEC often forms biofilms, complicating treatment and contributing to its persistence within the host [5,6]. Epidemiologically, EAEC has been implicated in both sporadic cases and outbreaks, affecting individuals across diverse age groups and geographical locations [7]. Its clinical relevance extends beyond acute gastroenteritis, as persistent infections have been associated with chronic sequelae, highlighting the need for a deeper understanding of the pathogen-host interactions [8,9].

EAEC, distinguished by its aggregative adherence and biofilm-forming capabilities, represents a significant public health concern globally [10,11]. The factors contributing to EAEC’s pathogenicity, including adhesion mechanisms and biofilm formation, play a crucial role in its ability to persist within the host’s gastrointestinal tract [11,12]. This persistence contributes to the global epidemiology of EAEC infections, with a notable impact on public health. Understanding these factors is essential for devising effective strategies to mitigate the burden of EAEC-related gastrointestinal illnesses. EAEC exhibits remarkable diversity in infection dynamics, adhesion mechanisms, IL-8 secretion, and inflammatory responses within the gastrointestinal tract [10,13]. Variances in virulence factors, antimicrobial resistance profiles, and genetic makeup contribute to distinct pathogenic potentials among EAEC strains [14,15]. This diversity intricately shapes the host–pathogen interface, influencing the severity and duration of inflammatory cascades through IL-8 secretion [16,17]. Understanding this multifaceted diversity is pivotal for tailoring targeted interventions, shedding light on the complex interactions governing EAEC infections, and offering insights for innovative strategies in gastrointestinal health maintenance.

The pivotal role of epithelial cells lining the gastrointestinal mucosa in sensing and responding to microbial challenges underscores the importance of understanding host defence mechanisms [18]. In instances of infection and inflammation, these cells deploy interleukin-8 (IL-8) as part of the host’s defence mechanism [19,20]. This study delves into the intricate interactions at the host–pathogen interface, with a specific focus on how EAEC influences IL-8 secretion and its subsequent impact on the host’s immune response. The focal point of this research is the complex relationship between EAEC and the inflammatory response within the gastrointestinal tract. The cascade triggered by EAEC infections involves the activation of various host factors, with IL-8 emerging as a pivotal player in this intricate process [20]. Functioning as a potent chemokine, IL-8 orchestrates the recruitment and activation of immune cells, thereby shaping the overall immune landscape within the gastrointestinal tract [21,22]. This dynamic interaction is crucial for deciphering the underlying mechanisms of EAEC pathogenesis, laying the groundwork for potential targeted therapeutic interventions [6].

In the intricate landscape of gastrointestinal infections, IL-8 takes centre stage as a central player, initiating an inflammatory response to combat invading pathogens [23,24]. Particularly within the context of EAEC infections, IL-8′s role becomes pronounced, influencing the severity and duration of the inflammatory cascade within the gastrointestinal tract [23,25]. The comprehensive exploration of EAEC pathogenicity, coupled with an in-depth analysis of its interaction with IL-8, yields valuable insights into the delicate balance governing gastrointestinal health. This understanding opens potential avenues for therapeutic interventions aimed at restoring equilibrium within the intricate host–pathogen interplay [23,25]. Beyond its role in infection, IL-8 holds significant implications for gastrointestinal health [26]. Its intricate involvement in maintaining mucosal integrity, immune surveillance, and overall homeostasis within the gastrointestinal mucosa emphasizes the paramount importance of a well-regulated IL-8 response [27,28]. Dysregulation of IL-8 levels is associated with pathological conditions within the gut, including inflammatory bowel diseases (IBD) and various gastrointestinal infections [29]. This study places a specific emphasis on IL-8 due to its relevance in the context of EAEC infections. The link between EAEC, IL-8 dysregulation, and the development of gastrointestinal pathologies provides a rationale for exploring targeted interventions. By modulating IL-8 responses, this research aims to contribute to the restoration of gastrointestinal health. The synergistic exploration of EAEC pathobiology and IL-8 dynamics offers a holistic understanding of the complex interplay shaping gastrointestinal health, paving the way for innovative strategies in disease prevention and intervention.

In this context, our study investigates several critical factors that influence IL-8 secretion and barrier integrity, including EAEC strain specificity, the impact of the Bacterial Infection Dose (BID), bacterial modes of infection, and the duration of infection (treatment time, TT). Our objective is to unravel the complex mechanisms underlying the crosstalk between EAEC-induced inflammation, LAB-mediated protection, and the maintenance of intestinal barrier function. This knowledge will not only advance our understanding of host–pathogen interactions but also shed light on the potential of LAB strains as therapeutic agents for gastrointestinal health.

## 2. Materials and Methods

### 2.1. Bacterial Strains and Culture Conditions

In this study, we employed two strains of diarrhoeagenic *E. coli*, namely 3591-87 and K2, alongside a non-diarrhoeagenic *E. coli* strain, (ND-EAEC) N23. These strains were isolated from unpasteurized fresh milk, except for 3591-87, which served as a positive clinical reference control [30,31] (Table 1). For our investigation, we selected two lactic acid bacteria (LAB) strains, *L. plantarum* FS2 and *P. pentosaceus* D39. These LAB strains exhibited promising probiotic characteristics and were chosen from a previous study involving the traditional non-alcoholic fermentation of maize to produce a West African gruel known as ‘Ogi’ [32]. Additional details regarding the LAB strains, including the probiotics used in this study and their respective culturing conditions, are detailed in Table 1.

### 2.2. Cell Culturing and Maintenance Conditions

We obtained human epithelial intestinal cells from colorectal adenocarcinoma, specifically Caco-2 cells (ATCC catalogue number, HTB-37, Manassas, VA, USA). These cells were maintained in Dulbecco’s modified Eagle medium (DMEM; Gibco, ThermoFisher, Waltham, MA, USA) supplemented with 4500 mg/L D-glucose, non-essential amino acids, and 110 mg/L of sodium pyruvate, as previously described [33].

To support their growth, the medium was further enriched with 10% (*v*/*v*) gamma-irradiated, heat-inactivated foetal bovine serum (FBS, Gibco, ThermoFisher, Waltham, MA, USA) and 1% penicillin-streptomycin. The cells were primarily cultured in T75 (75 cm^2^) cell culture flasks (catalogue number 658940, Greiner Bio-One GmbH, Frickhausen, Germany). Subculturing occurred at 60 to 70% confluence, with a 1:3 ratio, followed by incubation at 37 °C with 5% CO_2_ in a CO_2_ humidified environment (95% air) using an incubator (Healforce, HF 212UV, Hong Kong, China). Subculturing took place every 3–5 days after trypsinization with 0.5% trypsin-EDTA (ThermoFisher, Waltham, MA, USA).

The cell monolayers were considered polarized when their trans-epithelial electrical resistance (TEER) values measured at least 1000 Ω cm^2^. Therefore, for this study, we utilized polarized Caco-2 cell monolayers (PCC-2CMLs) with TEER values ranging from approximately 1000 to 2000 Ω cm^2^, based on previous research [34]. Monolayers with TEER values below 1000 Ω cm^2^ were excluded due to the potential for increased permeability.

We employed Caco-2 cells from passages 30–39 for all experiments, ensuring that the cell cultures were routinely examined and confirmed to be free of bacterial and mycoplasma contamination before each use. At least two hours prior to conducting various experiments, the cells were nourished with serum- and antibiotic-free medium.

### 2.3. Preparation of Epithelial Cells for Inflammation Assays

Caco-2 cells were seeded at a density of 5.0 × 10^5^ cells/1.12 cm^2^ using Corning 12-well plates with sterile Coaster Snapwell collagen-coated filter inserts (Transwell^®^-COL, 12 mm diameter, 1.12 cm^2^ cell growth area, 0.4 µm pore with CN, 3493, Corning B.V., Glendale, AZ, USA). After seeding, the cells underwent a 21-day differentiation process. In preparation for bacterial infection, PCC-2CMLs were washed with phosphate-buffered saline (PBS) three times to remove foetal bovine serum (FBS) and antibiotics. Approximately 2 h prior to infection, freshly prepared DMEM (contained 4500 mg/L D-glucose, non-essential amino acids, and 110 mg/L sodium pyruvate but without FBS and antibiotics) was added to both the apical and basolateral compartments of the wells, with and without epithelial cells facilitating the subsequent experiments.

### 2.4. The Effects of EAEC and LAB on Epithelial Barrier Integrity

In this study, we assessed the ability of selected LAB strains to protect and maintain epithelial-like PCC-2CMLs from the damaging effects of diarrhoeagenic EAEC. Briefly, 18 h old bacterial cultures were standardized [6.0 × 10^8^ CFU/mL for EAEC and 6.0 × 10^9^ CFU/mL for LAB (Table 1)] using PBS and a McFarland densitometer (DEN-1 Model, Grant bio, Sia Biosan, Riga, Latvia). These strains were then homogenized in serum- and antibiotic-free DMEM to achieve final bacterial densities of (EAEC, 6.0 × 10^7^ and LAB, 6.0 × 10^8^ CFU/mL).

Selected wells containing PCC-2CMLs were either monoinfected by replacing the cell culture medium in the apical chambers with 25 µL of DMEM-bacterial suspension and PBS or coinfected with combinations of 25 µL each of an EAEC strain (3591-87, K2, and N23) and a LAB strain (*B. bifidum* ATCC 11863, *L. plantarum* FS2, and *P. pentosaceus* D39, as listed in Table 2). These challenged PCC-2CMLs were then incubated at 37 °C in the presence of 5% CO_2_ for 6 h. We measured the initial and final resistance across the PCC-2CMLs to determine TEER. Additionally, 25 µL of supernatants from the apical chamber were harvested and stored at −20 °C for subsequent IL-8 assay [34].

### 2.5. Effect of the Bacterial Infection Dose (BID) on Epithelial Barrier Integrity

We monitored the effect of the D-EAEC K2 infection dose on TEER and IL-8 induction as previously reported [35] with a few modifications. Briefly, EAEC K2 cultures (18 h old) were standardized (1.5 × 10^9^ CFU/mL) as previously described (Section 2.2). The bacterial cells were homogenized with serum- and antibiotic-free DMEM to a final concentration of (1.5 × 10^8^ CFU/mL). This bacterial cell-culture medium suspension was taken through ten-fold serial dilutions with further homogenizations to obtain various bacterial concentrations up to 1.5 × 10^2^ CFU/mL and then used to challenge PCC-2CMNLs by replacing the serum- and antibiotic-free DMEM with the bacterial suspension followed by incubation (Section 2.2). The initial and final resistance measurements were taken across the PCC-2CMLs for the estimation of TEER, whilst supernatants were harvested and kept (−20 °C) for IL-8 assay as previously described [34].

### 2.6. Bacterial Infection Mode and Treatment Time (TT) Effects on Epithelial Barrier Integrity

This assay was carried out to determine the effect of TT and bacterial infection mode; 1. Simultaneously with EAEC and LAB; 2. With EAEC an hour before LAB; or 3. LAB an hour before EAEC on TEER and IL-8 of the PCC-2CMNLs termed bacterial competitive exclusion from adhesion assay (BCEFAA), bacterial displacement from adhesion assay (BDFAA), and bacterial inhibition from adhesion assay (BIFAA), respectively. This study was restricted to EAEC K2 and *L. plantarum* FS2. Bacterial cultures (18 h old) were standardized (1.5 × 10^8^ and 1.5 × 10^9^ CFU/mL) (Section 2.2) and used to infect PCC-2CMNLs in different modes [33] resulting in final BIDs of 7.5 × 10^7^ and 7.5 × 10^8^ CFU/well for EAEC and LAB, respectively. The challenged PCC-2CMNLs were incubated (37, 5% CO_2_) and assessed for their initial and final TEERs (at 4, 8, 12, 16, 20, 24 and 28 h) whilst their corresponding supernatants were collected from the apical chambers and stored (−20 °C) for IL-8 assay as previously described [34].

### 2.7. Bacterial Mode of Infection and TT Effects on Adhesion

This study aimed to determine the effects of TT and different modes of infection on the competence of bacterial adhesion to PCC-2CMNLs. The selected bacterial cultures (EAEC K2 and *L. plantarum* FS2, 18 h old) were standardized as described earlier (Section 2.2). The PCC-2CMNLs were mono- and coinfected in different modes with the selected bacteria [33] and resulting final bacterial infection densities (Section 2.2). The experimental setups were then incubated (Section 2.2). The bacterial cells (EAEC and LAB) were evaluated for their competence for adhesion to the monolayers (2, 4, 8, 12, 16, 20, 24 and 28 h).

### 2.8. Interleukin 8 (IL-8) Assay

A commercially available sandwich Enzyme Linked-Immunosorbent Assay (ELISA) kits (Elabscience Biotechnology Inc., Houston, TX, USA) (CN, E-EL-H0048) was purchased and used to evaluate IL-8 strictly according to the manufacturer’s instructions. Briefly, anti-human IL-8 pre-coated 96 well strip plates were individually treated with serially diluted reference standards and then incubated [room temperature (RT), 1 h]. The plates were washed (3×) with PBS followed by treatment of each well with biotinylated antibody reagent and incubation (RT, 1 h). The plates were washed again (3×) followed by treatment with 100 μL of streptavidin-horseradish peroxidase (HRP) solution, covered with Petri film and then incubated (25 °C, 30 min). This was followed by washing (3×) and then treatment of each well with 100 μL of TMB (3,3′,5,5′-tetramethylbenzidine). The plates were further incubated (RT, 30 min, dark room). Each well was finally treated with 100 μL of the stop solution to terminate the reactions. The optical density readings (500 nm) of the plates were measured using a filter-based multi-mode microplate reader (FLUOstar Omega, BMG LabTech, Ortenberg, Germany). The entire experiment was repeated by replacing the serially diluted reference standards with the thawed and preincubated (RT, 15 min) harvested supernatants. The IL-8 concentrations of the samples were calculated with reference to the linear equation generated from the optical densities of the reference standards.

### 2.9. Trans-Epithelial Electrical Resistance (TEER) Assay

This assay was carried out by following previously laid down protocols [34] with a few modifications. The electrical resistance was measured across the monolayers (from the apical to the basolateral sides) using a Millicell ERS-2 electrode (MERSSTX01) volts/ohmmeter resistance system (Millipore Corporation, Bedford, MA, USA). To obtain the true resistance values, the background resistance for the cell culture membrane inserts with (the serum- and antibiotic-free) medium was subtracted from the initial and final resistance readings. The TEER value was obtained as a product of the resistance value and the cell culture insert membrane area in cm^2^.

### 2.10. Statistical Analysis

All experiments were independently conducted in triplicates. Except for the analysis of the effects of bacterial treatment and TT on bacterial adhesion and TEER using a two-way analysis of variance (ANOVA), all other data were subjected to analysis using the one-way ANOVA tool-pack of Statgraphics Centurion XIX [36]. Results were compared at a 95% confidence level, and mean values with *p* ≤ 0.05 were deemed statistically significant. Multiple range tests were performed using Fisher’s least significant difference. The obtained data was primarily utilized to create bar charts, visually representing the statistical findings. Correlation analysis was conducted by pairing variables, including bacterial adhesion, TEER and IL-8 secretion from the Caco-2 monolayers, at a 95% confidence level using the correlation analysis tool-pack of Statgraphics Centurion XIX [36].

## 3. Results and Discussion

### 3.1. Cytokine Secretion from Caco-2 Monolayers in the Presence or Absence of EAEC and LAB

Our results demonstrate that both EAEC and LAB induce the secretion of the proinflammatory cytokine interleukin-8 (IL-8) from PCC-2CMNLs. However, the amount of IL-8 induced by LAB strains was significantly lower (*p* < 0.05) than that induced by EAEC strains (Figure 1).

In monoinfections with EAEC strains, D-EAEC K2 induced the highest level of IL-8 secretion, followed by the clinical positive reference D-EAEC 3591-87, and then ND-EAEC N23, resulting in fold increases of 14.5, 13.3, and 4.2, respectively, compared to control setups. These results aligned with findings by [33] during which D-EAEC K2 and ND-EAEC N23 exhibited the highest and the lowest adhesion scores, respectively.

The selected LAB strains exhibited varying degrees of mitigation against the three EAEC strains in their ability to induce IL-8 secretion, depending on strain specificity. Particularly, against the clinical D-EAEC reference strain 3591-87, *P. pentosaceus* D39 reduced IL-8 induction the most, followed by *B. bifidum* ATCC 11863, and then *L. plantarum* FS2, with reductions of 198.53 pg/mL (52.4%), 184.9 pg/mL (48.8%), and 145.8 pg/mL (38.5%), respectively.

Concerning D-EAEC K2, *B. bifidum* ATCC 11863 demonstrated the highest mitigation of IL-8 secretion, followed by *L. plantarum* FS2, and then *P. pentosaceus* D39, with reductions of 265.6, 230.2, and 203.7 pg/mL, corresponding to 64.4%, 55.6%, and 49.4%, respectively. *L. plantarum* FS2 showed the most significant mitigation against ND-EAEC N23, followed by *P. pentosaceus* D39, and then *B. bifidum* ATCC 11863, resulting in reductions of 63.3, 48.6, and 39.5 pg/mL, signifying 53.5%, 41.3%, and 33.4%, respectively.

These results were unexpected, as we anticipated *L. plantarum* FS2 to have the highest mitigatory effect against the selected EAEC strains. This is because this LAB strain exhibited the highest adhesion ability to the PCC-2CMNLs and demonstrated excellent competitive exclusion, displacement, and inhibitory abilities against most EAEC strains [33].

Numerous studies have shown that various cytokines regulate intercellular tight junctions, cytoskeletal structure, and function [37]. IL-8 is a well-known proinflammatory cytokine that recruits neutrophils, antigen-presenting cells, and other immune cells to sites of tissue injury or infection. It has also been associated with pathogen-induced alterations in intercellular tight junctions [38].

Our current study reveals that, except for ND-EAEC N23, the other two D-EAEC strains were associated with significant increases in IL-8 secretion. In contrast, the selected LAB strains were mostly associated with lower levels of IL-8 expression, suggesting the potential for mitigating intestinal epithelial inflammation while preserving epithelial barrier integrity and function.

Our results indicate that monoinfection with the pathogens, EAEC 3591-87 and K2, led to higher levels of inflammation (IL-8 secretion) than coinfections with the selected LAB strains. These findings were consistent with [39,40]. In contrast, inflammation due to monoinfection with LAB was significantly lower (*p* < 0.05) than that with the pathogens agreeing with results from [39].

Earlier studies have indicated that the levels of induced IL-8 moderately correlate with endothelial and epithelial permeability, suggesting that IL-8 can be a reliable in vitro biomarker for assessing the severity of inflammatory-related illnesses. However, it is worth noting that IL-8 may not immediately reflect changes in endothelial/epithelial permeability and may take up to two days to significantly affect the permeability of the model [39]. This delay is attributed to the low levels of secreted IL-8, which predominantly occur during the first post-infection day.

### 3.2. BID Effect on IL-8 Induction

The results regarding the effect of the bacterial infection dose (BID) on EAEC K2′s ability to induce IL-8 from the differentiated Caco-2 cell monolayers (DCC-2CMLs) indicate a dose-dependent relationship (Figure 2). IL-8 secretion increased from the control setups to those treated with a final bacterial concentration of 3.7 log10 (CFU/well), resulting in an increase of 208.4 pg/mL (4.1-fold). Subsequently, the IL-8 induction ability of this D-EAEC strain continued to increase, reaching 262.4, 309.4, 315.4, 325.5, 345.8, and 360.3 pg/mL, signifying 4.9, 5.6, 5.7, 5.9, 6.2, and 6.4-fold increases relative to the controls for the DCC-2CMLs infected with 4.7, 5.7, 6.7, 7.7, 8.7, and 9.7 log10 (CFU/well), respectively.

The adhesion and colonization of the gut by enteropathogens stimulate the host’s innate inflammatory response, leading to the secretion of IL-8 and other pro-inflammatory substances. This, in turn, attracts neutrophils and other inflammatory cells to the site of infection. However, prolonged, and excessive neutrophil infiltration can lead to persistent inflammation, ultimately resulting in cell damage, deterioration of epithelial barrier function, and the onset of diarrhoea.

Our results clearly demonstrate that the selected LAB strains did not trigger IL-8 secretion from the PCC-2CMLs. Moreover, a LAB dose of approximately 10^8^ CFU/mL proved effective in preventing IL-8 secretion by the PCC-2CMLs, consistent with prior research findings [35]. These results suggest that the selected LAB strains may have the potential to prevent enteropathogens from inducing IL-8 secretion.

It is worth noting that enteropathogen-induced gut inflammation can alter the composition and stability of the gut microbiome, disrupt colonization resistance, and promote the proliferation of pathogens within the gut [41]. The fact that *L. plantarum* FS2 and *P. pentosaceus* D39 competitively excluded, displaced, and inhibited D-EAEC from the intestinal epithelium suggests that they could be promising candidates for the development of functional foods. Per our expectations however, our findings presented a divergent perspective compared to the data reported in prior studies [42]. Specifically, our results revealed a direct correlation between the BID and the IL-8 response of intestinal epithelial cells.

Our results align with our expectations, demonstrating a proportional relationship between the BID and the IL-8 response of intestinal epithelial cells. As the BID increases, the adherence of enteropathogens to the epithelial cells also increases, leading to elevated secretion of the proinflammatory cytokine, consistent with previous findings [35].

### 3.3. Bacterial Infection Mode and TT Effects on IL-8 Secretion

Our findings (Figure 3), reveal a significant impact of TT (incubation time) on EAEC K2′s ability to induce the secretion of IL-8 in contrast to untreated DCC-2CMLs and those treated with *L. plantarum* FS2 (*p* < 0.05). Unlike LAB, the IL-8 induction ability of EAEC K2 displayed a progressive increment after the 8th, 12th, 16th, 20th, 24th, and 28th hours, by 0.5, 1.2, 2.1, 2.4, 3.0, and 3.3-fold, respectively.

Results from the BCEFAA showed continuous increments from the 4th to the 8th and 12th hours, by 0.4 and 0.9-fold, respectively compared to the 4th hour, after which the rate of increment declined from the 16th hour onwards, with values of 0.7, 0.6, 0.4, and 0.3-fold relative to the 4th hour, respectively. Similarly, results from the BDFAA demonstrated increments in IL-8 secretion from the 4th to the 12th hour, with an additional 0.7-fold compared to the 4th hour. This increase gradually reduced up to the 28th hour, with an additional 0.3-fold relative to the 4th hour. Likewise, results from the BIFAA showed a gradual increase in IL-8 secretion up to the 12th hour, by 0.7-fold, which subsequently reduced up to the 28th hour, by 0.4-fold compared to the 4th hour.

Our results indicate that from the 4th to the 12th hour, both BCEFAA and BDFAA did not reduce but rather increased IL-8 secretion due to pathogen virulence. This observation could be attributed to the fact that the reduction in IL-8 induction required a relatively longer period. These two modes of infection eventually led to a reduction in IL-8 secretion from the 16th to the 28th hour.

The IL-8 induction pattern during BIFAA differed from the two previous modes of bacterial infection. No difference was observed between the setup for infection of the Caco-2 monolayers with pure EAEC K2 alone and those infected with *Lactobacillus plantarum* FS2, followed by EAEC K2 during BIFAA after the 8th hour. This suggests that the prior infection of DCC-2CMLs with LAB before EAEC K2 could not confer any protection against pathogen virulence. From the 12th to the 28th hour, this infection mode consistently reduced the IL-8 secretion effect of EAEC K2.

Our results showed significant reductions in epithelial barrier function caused by enteropathogens’ virulence, coupled with IL-8 secretion mainly based on BID and TT which further support findings from earlier studies [35,43]. Studies involving coinfection, such as *Anaplasma phagocytophilum* and *B. burgdorferi*, have shown that secreted IL-8 levels correspond to tight junction formation (TEER), suggesting IL-8′s role as a biomarker for assessing the severity of infections [44] Disruption of intercellular tight junction proteins and increased endothelial permeability have been linked to higher IL-8 secretion [45]. Consequently, higher severity, as indicated by increased endothelial permeability, may be associated with elevated IL-8 levels, as demonstrated in monoinfection with EAEC compared to coinfection with the selected LAB.

In a separate study on HIV infection, higher IL-8 levels were associated with enhanced viral replication [46]. Moreover, increased endothelial permeability correlated with higher viral dose [47], also supporting the findings of this study.

### 3.4. Effects of Bacterial Monoinfection on TEER

The results obtained from monoinfected DCC-2CMLs revealed distinct effects on TEER. *B. bifidum* ATCC 11863 caused a 6.2% reduction in TEER compared to the control setups (non-infected DCC-2CMLs), decreasing from 104.6% to 98.5%. Similarly, monoinfection with *L. plantarum* FS2 and *P. pentosaceus* D39 led to a TEER deterioration of 9.0% and 7.3%, bringing TEER values down to 95.6% and 97.3%, respectively from the experimental control (Figure 4).

Among the EAEC strains, D-EAEC K2 significantly decreased TEER scores by 46.0 to 58.6% (*p* < 0.05). Following this, the clinical positive reference strain D-EAEC 3591-87 exhibited a decrease of 39.1%, resulting in a TEER of 65.5%, and ND-EAEC N23 showed the least reduction of 18.7%, with a TEER of 86.0%. Our findings for both EAEC and LAB strains underscored the strain-specific effects, aligning with prior studies [33].

The measurement of TEER plays a pivotal role in assessing structural and functional maintenance, directly linked to epithelial barrier integrity and permeability [35,48]. TEER measurements rely on both cellular and shunt resistances, which operate in parallel. Interestingly, none of the three LAB strains demonstrated any detrimental effects; instead, they improved and maintained epithelial barrier integrity, consistent with earlier research findings [35,49].

In a separate study, certain LAB strains, such as *Lactobacillus rhamnosus* HN001, *L. rhamnosus* L34, *Lactobacillus acidophilus*, and *L. plantarum*, were reported to significantly reduce pathogen virulence [50]. On the other hand, LAB strains like *L. plantarum* and *L. rhamnosus* were shown to stimulate the host immune response [51,52]. Our current study indicates that the selected LAB strains have a positive impact on improving intercellular tight junctions.

### 3.5. Effects of LAB and EAEC Coinfection on TEER

The co-infection of the EAEC with the three selected LAB strains demonstrated varying abilities of the latter in ameliorating TEER levels, which had been adversely affected by the deteriorative effects of EAEC on the PCC-2CMLs. After a six-hour treatment period, PCC-2CMLs, previously monoinfected with D-EAEC K2, displayed the most substantial TEER restoration. Notably, *P. pentosaceus* D39 led this restoration, achieving a remarkable improvement of 30.2%, with TEER levels increasing from 58.6% to 88.8% (Figure 4).

*L. plantarum* FS2 also exhibited notable competence in countering the effects of D-EAEC K2, resulting in a TEER increase of 25.8%, reaching 84.4%. Additionally, *P. pentosaceus* D39 effectively restored TEER levels disrupted by D-EAEC 3591-87, showing a substantial 22.2% increase, reaching 87.7%. Furthermore, *B. bifidum* ATCC 11863 demonstrated its efficacy by achieving an 18.8% TEER increase when countering the effects of D-EAEC K2, with levels reaching 77.8%. Lastly, *B. bifidum* ATCC 11863 also exhibited an 18.8% TEER increase when combating D-EAEC 3591-87, resulting in a TEER value of 84.4%.

The reductions in TEER, leading to cellular structural damage and cytokine induction, exhibited variations among bacterial species and strains, irrespective of pathogenicity. Notably, scanning electron micrographs showed that coculturing PCC-2CMLs with EAEC for 18 h predominantly deteriorated the integral structure of the Caco-2 cells (Figure 5). Specifically, D-EAEC 3591-87 and K2 had a more pronounced impact on reducing TEER in PCC-2CMLs compared to their non-diarrheagenic counterpart, EAEC N23, as well as the LAB strains (as observed in Figure 4).

In our current study, PCC-2CMLs underwent challenges by EAEC in the presence or absence of different adhering LAB strains. At the end of the treatment, unchallenged monolayers maintained their TEER values, thus preserving their intestinal barrier integrity, consistent with reported findings [35]. This observation was consistent with the results obtained for PCC-2CMLs treated with the selected LAB. In these cases, the TEER values for the PCC-2CMLs remained relatively stable, highlighting the LAB’s ability to uphold intercellular barrier integrity and function [53,54]. However, when PCC-2CMLs were challenged with D-EAEC, reductions in TEER were observed, as previously reported [35,55]. These reductions in TEER indicate deteriorations in intestinal barrier integrity and demonstrate some strain-dependent effects, aligning with prior studies [35,56].

Interestingly, the three LAB strains, when cocultured independently with PCC-2CMLs over the 6 h incubation period, did not appear to significantly influence TEER levels. It’s worth noting that reductions in TEER can result from cytotoxic pore formation in cells, but this phenomenon also depends on the physiological regulation of intercellular tight junctions [57,58]. These tight junctions are primarily maintained by proteins such as claudins, occludins, and zonal occludins (ZO), including ZO-1, ZO-2, and ZO-3 proteins [59,60].

### 3.6. BID Effect on TEER

The impact of the bacterial infection dose (BID) on TEER was assessed using EAEC K2 as the model organism. The results depicted in Figure 6 illustrate an exponential decline in TEER as the BID increases.

Uninfected PCC-2CMLs exhibited the highest TEER value, indicating the integrity of intercellular tight junctions, which are essential for maintaining epithelial barrier integrity and function (97.3%). However, upon exposure to a BID of 0.36 log10/well, the epithelial barrier’s integrity, represented by TEER, experienced a significant (*p* < 0.05) reduction of 20.8%, resulting in a TEER value of 76.4%. This deterioration continued as the BID increased to 1.36 log10/well, leading to a further decline in TEER by 38.7%, resulting in a TEER value of 58.5%.

The trend persisted until the epithelial barrier’s integrity reached a drastic reduction by 95.2%, culminating in a final TEER value of only 2.0%. This study unequivocally demonstrates the direct relationship between the BID and epithelial barrier integrity (TEER), with higher BID values corresponding to lower TEER values and compromised intercellular tight junctions.

As previously reported [61,62], bacterial adhesion tends to increase with the infection dose. Notably, adherent bacteria like the EAEC strain utilized in our study predominantly attach to the apical sides of the epithelium [61,63]. Consequently, the variation in TEER induction with BID aligns with expectations, as earlier reported [35,54].

### 3.7. Bacterial Mode of Infection and TT Effects on TEER

Our results reveal that PCC-2CMLs, which were monoinfected with D-EAEC K2, experienced a significant deterioration in their intercellular tight junctions from the 4th (61.4%) to the 28th hour (4.6%) of treatment, representing deviations from their controls ranging from 40.0% to 104.5%, respectively (Figure 7). The various modes of coinfecting the intestinal epithelium showed that infecting PCC-2CMLs with *L. plantarum* FS2 an hour before introducing D-EAEC K2 (Figure 7) was the most effective approach for maintaining intercellular tight junctions. This was followed by simultaneous coinfection of the two bacteria. Conversely, pre-infecting the monolayers with D-EAEC K2 an hour before introducing *L. plantarum* FS2 was generally the least effective in ameliorating the epithelial barrier. These results were not surprising because with simultaneous coinfection, the LAB was introduced to counter the EAEC right from the start. Moreover, when the monolayers were infected with the LAB an hour before the pathogen, the former could initiate some prophylactic processes before the introduction of the latter. Consequently, when the pathogen was first introduced, the therapeutic capacity of the LAB might have been delayed compared to the other two modes of infection.

The current study further substantiates the mechanisms by which probiotics exert their beneficial effects while emphasizing the competence of the selected LAB in safeguarding polarized epithelial cells against the deleterious effects of diarrhoeagenic *E. coli* at various levels. Additionally, our results illustrate that treatment with the selected LAB mitigated the detrimental effects of the D-EAEC strain on the epithelial barrier integrity and function. Furthermore, we demonstrated that the selected D-EAEC strains led to reductions in TEER, potentially increasing epithelial permeability. These findings imply that the selected LAB strains may be valuable for protecting and maintaining intercellular tight junctions and epithelial barrier integrity, aligning with earlier reports [64].

While various Lactobacillus strains have been reported in clinical studies to confer beneficial health effects on their hosts through mechanisms such as bactericidal or bacteriostatic agents [65], regulation of immunomodulatory effects [66], or competitive exclusion of pathogens [33], their precise mechanisms of action remain unclear. The current findings suggest that *L. plantarum* FS2, *P. pentosaceus* D39, and *B. bifidum* ATCC 11863 can maintain intestinal barrier functions to varying degrees by preventing disruptions caused by enteropathogens. This is achieved through the upregulation of TEER in PCC-2CMLs and the downregulation of permeability by limiting the secretion of inflammatory cytokines, among other mechanisms, as earlier reported [67,68].

### 3.8. Bacterial Mode of Infection and TT Effects on TEER

In this study, we investigated the dynamic interactions between probiotic strains and pathogenic bacteria within the context of gastrointestinal health. Specifically, we explored the adhesion behaviour of D-EAEC K2 and *L. plantarum* FS2 to DCC-2CMLs over varying durations of exposure, shedding light on the temporal aspect of bacterial adhesion. Gastrointestinal infections are often linked to disruptions in the composition and function of the gut microbiome, making probiotics crucial players in maintaining human health, especially in the era of antibiotic-resistant pathogens. These experiments were designed with the objective of offering valuable insights into the probiotics’ abilities in competitive exclusion, displacement, and inhibition when confronted with the selected EAEC strains.

While the adhesion of D-EAEC K2 to DCC-2CMLs was significantly greater (*p* < 0.05) than that of *L. plantarum* FS2 at specific time points (12 and 24 h), both strains exhibited progressive, statistically significant increases in adhesion with increasing exposure time (TT), ranging from 25.6% to 73.2% for D-EAEC K2 and 24.3% to 70.9% for *L. plantarum* FS2 (Figure 8). This observation highlights the time-dependent nature of bacterial adhesion. Gastrointestinal infections often result from imbalances in the composition and function of the human gut microbiome [69,70]. Probiotics play a crucial role in shaping the gut microbiome and contributing to overall human health. In the face of antibiotic-resistant pathogens [71], novel treatments and preventive techniques like probiotics have become indispensable. Our results demonstrate that the *L. plantarum* FS2 have varying abilities in competitive exclusion, displacement, and inhibition against the selected EAEC strains, confirming earlier research findings [33,72].

In our study, DCC-2CMLs were infected by the two antagonistic bacteria, D-EAEC K2 and *L. plantarum* FS2, using three different infection modes: BCEFAA, BDFAA, and BIFAA. *L. plantarum* FS2 consistently outcompeted EAEC K2 in adhesion (*p* < 0.05), with a progressive increase from 6.1% to 44.0% over 2 to 28 h (Figure 8).

Additionally, *L. plantarum* FS2 progressively displaced EAEC K2 with TT. At 2, 4, 8, 12, 16, 20, 24, and 28 h, LAB displaced EAEC K2 by 5.5, 20.7, 16.9, 24.8, 24.8, 34.2, 40.0, and 54.0%, respectively. Surprisingly, LAB’s displacement ability exceeded their competitive exclusion capacity, contrary to our expectations, suggesting that in the absence of competing LAB, EAEC cells adhered more slowly to epithelial cells, as previously confirmed [33].

Pre-infecting DCC-2CMLs with *L. plantarum* FS2 an hour before EAEC K2 infection progressively inhibited EAEC K2 adhesion. LAB inhibited EAEC K2 by 7.7, 14.7, 20.6, 29.6, 32.9, 39.0, 40.7, and 60.4% at 2, 4, 8, 12, 16, 20, 24, and 28 h, respectively. Remarkably, the degree of inhibition of EAEC K2 adhesion consistently exceeded competitive exclusion and displacement, aligning with prior research [73].

Furthermore, our results (Figure 9) suggest that, unlike D-EAEC K2, *L. plantarum* FS2 did not exhibit a correlation between adhesion and TEER variables (*R* = 0.6440; *p* > 0.05), partially aligning with earlier studies [74]. Interestingly, both bacteria demonstrated a linear relationship between adhesion and IL-8 secretion, with strong positive correlations between their adhesion abilities and IL-8 secretion (*R* = 0.9552; *p* < 0.05) and (*R* = 0.9546; *p <* 0.05) for EAEC K2 and *L. plantarum* FS2, respectively (Figure 9). These findings disagreed with a previous report [75]. Additionally, both EAEC K2 and *L. plantarum* FS2 induced significant IL-8 secretion when TEER values were low and vice versa, as indicated by strong negative correlations (*R* = 0.9740; *p* < 0.05) and (*R* = 0.7906; *p* < 0.05) for EAEC K2 and *L. plantarum* FS2, respectively (Figure 9). Notably, no prior study has reported a correlation between IL-8 induction and TEER to our knowledge.

The results and discussions in this study illuminate crucial insights into the intricate interplay between EAEC, specific LAB, and the host intestinal environment. Notably, both EAEC and LAB induce IL-8, emphasizing their impact on intestinal inflammation. The study unveils strain-specific variations in LAB’s ability to mitigate IL-8 secretion induced by different EAEC strains, with *L. plantarum* FS2 emerging as a standout candidate. These findings carry significant clinical implications, suggesting the potential of certain LAB strains, especially *L. plantarum* FS2, in mitigating intestinal inflammation and preserving epithelial barrier integrity. Moreover, the study delves into the dynamics of the BID and TT on IL-8 induction and epithelial barrier integrity. The results establish a direct relationship between the BID and IL-8 secretion, providing insights into the importance of controlling bacterial load in infections. The time-dependent nature of adhesion between EAEC and LAB, with *L. plantarum* FS2 exhibiting superior inhibition of EAEC adhesion, adds a layer of complexity to understanding these interactions. The observed correlations between adhesion, IL-8 secretion, and barrier integrity further contribute to the nuanced understanding of gut microbiome dynamics. Particularly, the strong positive correlation between adhesion abilities and IL-8 secretion for both EAEC and *L. plantarum* FS2 underscores the intricate relationship between bacterial interactions and host immune responses.

The present study, much like many others, is not exempt from limitations and biases. The study reveals strain-specific effects of LAB in mitigating IL-8 secretion, which may limit the generalizability of the findings. Future studies should consider exploring a broader spectrum of LAB strains to establish a comprehensive understanding of their potential applications. While IL-8 serves as a reliable biomarker for inflammation, relying solely on this cytokine may overlook the broader inflammatory landscape. Exploring additional cytokines and markers related to gut health could provide a more comprehensive picture. The study utilizes an in vitro model (DCC-2CMLs) to simulate interactions between bacteria and epithelial cells. Translating these findings to in vivo scenarios requires caution due to the inherent differences in the complexity of the gut environment.

The study references prior research to support certain expectations, potentially introducing publication bias. A systematic review approach could mitigate this bias by considering a broader range of relevant literature. The selection of specific EAEC and LAB strains might introduce bias based on their inherent characteristics. A more randomized selection of strains could enhance the study’s objectivity.

Future research activities need to consider investigating a wider array of LAB strains to identify the most effective strains in mitigating IL-8 secretion and maintaining barrier integrity. In vivo studies need to be conducted to validate the observed effects in a more complex and dynamic gut environment, to provide a more accurate representation of potential outcomes. A multi-marker approach needs to be explored by assessing various cytokines and biomarkers related to gut health to capture a more nuanced understanding of the inflammatory response. The long-term effects of LAB supplementation on gut health need to be investigated, considering factors such as gut microbiome composition and overall host well-being. The gap between laboratory findings and clinical relevance needs to be bridged by conducting studies that directly assess the impact of LAB supplementation on individuals with gastrointestinal conditions.

While the current study contributes valuable insights, addressing these limitations and biases, along with exploring new avenues for research, will enhance the robustness and applicability of the findings in advancing our understanding of the complex interactions within the gut microbiome.

## 4. Conclusions

In conclusion, the study investigated interactions between enteroaggregative *E. coli* (EAEC) and specific lactic acid bacteria (LAB) in relation to cytokine secretion, barrier integrity, and bacterial adhesion. The results indicate that EAEC induces the proinflammatory cytokine IL-8, suggesting a role in intestinal inflammation. Certain LAB strains, especially *L. plantarum* FS2, demonstrated several potentials in reducing IL-8 secretion and preserving barrier integrity, making them promising candidates for promoting gastrointestinal health. This study emphasized the importance of controlling bacterial load in infections and highlighted time-dependent aspects of adhesion. *L. plantarum* FS2 showed exceptional ability to inhibit EAEC adhesion, indicating its potential in preventing infections. Complex relationships between adhesion, cytokine secretion, and barrier integrity were observed, suggesting avenues for further research. Overall, the current study advances our understanding of gut microbiome dynamics and offers insights into using probiotics for improving gastrointestinal health.

## Figures and Tables

**Figure 1 microorganisms-11-02942-f001:**
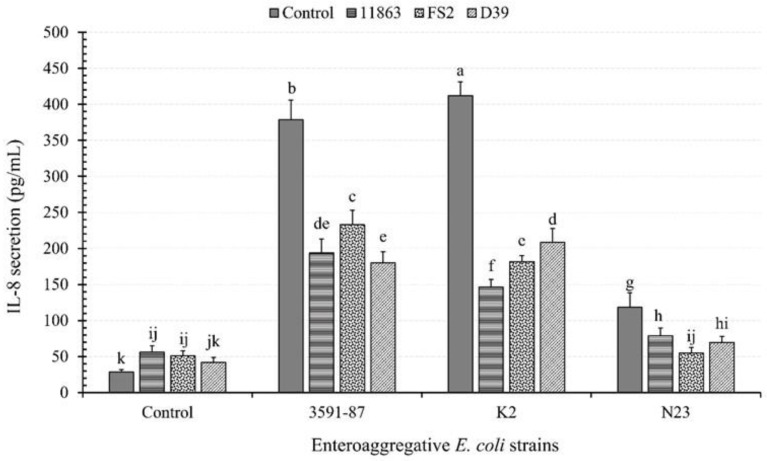
The effect of enteroaggregative *E. coli* (EAEC) and lactic acid bacteria (LAB) on the induction of interleukin 8 (IL-8) from polarized Caco-2 cell monolayers. The EAEC include diarrhoeagenic 3591-87 and K2 and a non-diarrhoeagenic (N23) strains. *B. bifidum;* ATCC, 11863 *L. plantarum*, FS2 and *P. pentosaceus*, D39 constitute the lactic acid bacteria (LAB) strains. Each bar is a mean of two independent replicates (*n* = 4) with its corresponding standard deviation. Bars with different letters (a–k) indicate significant differences (*p* < 0.05) according to Fisher’s LSD test.

**Figure 2 microorganisms-11-02942-f002:**
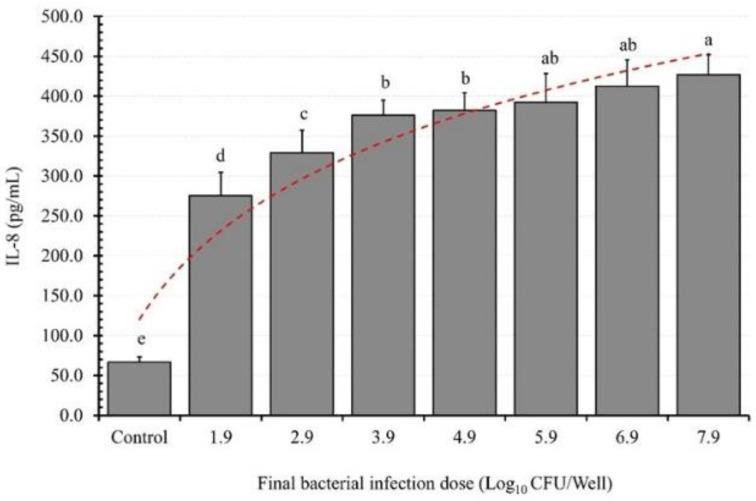
The effect of bacterial (enteroaggregative *E. coli*, EAEC, K2) infection dose on IL-8 induction from polarized Caco-2 cell monolayers. Each bar is a mean of two independent replicates (*n* = 4) with its corresponding standard deviation. Bars with different letters (a–e) indicate significant differences (*p* < 0.05) according to Fisher’s LSD test.

**Figure 3 microorganisms-11-02942-f003:**
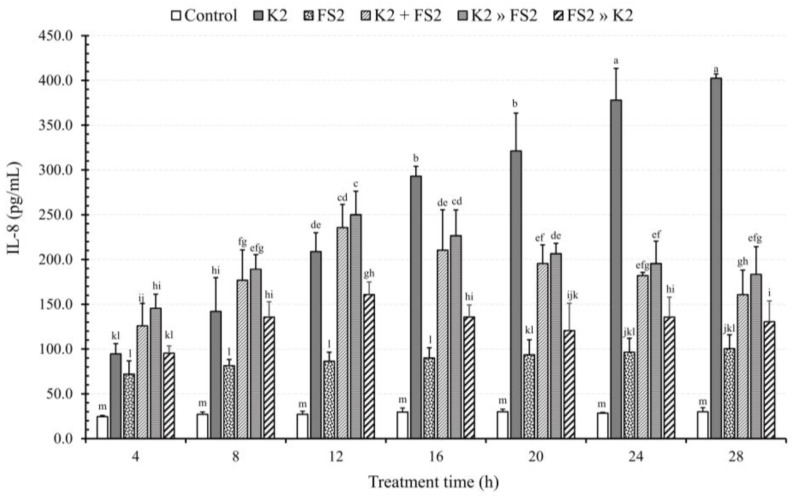
The effect of bacterial (enteroaggregative *E. coli*, EAEC, K2 and *L. plantarum*, FS2) infection mode and treatment time on IL-8 induction from polarised Caco-2 cell monolayers. Each bar is a mean of two independent replicates (*n* = 4) with its corresponding standard deviation. Bars with different letters (a–m) indicate significant differences (*p* < 0.05) according to Fisher’s LSD test.

**Figure 4 microorganisms-11-02942-f004:**
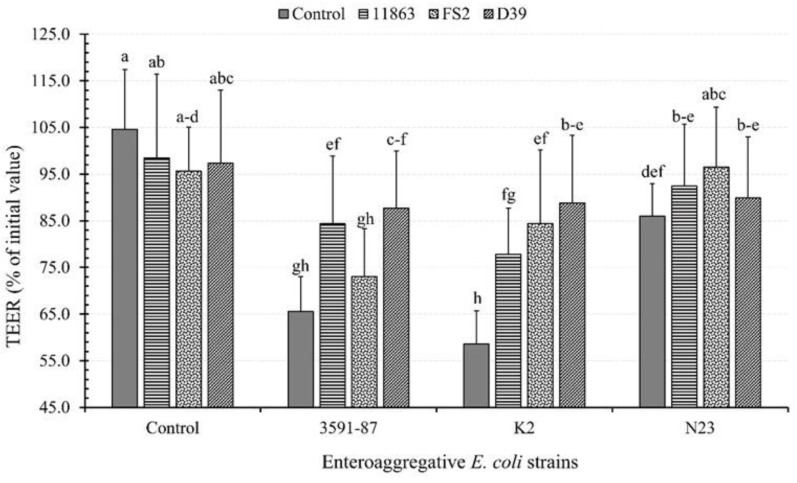
The effect of enteroaggregative *E. coli* (EAEC) and lactic acid bacteria (LAB) on transepithelial electrical resistance (TEER) of polarised Caco-2 cell monolayers. The EAEC include diarrhoeagenic 3591-87 and K2 and a non-diarrhoeagenic (N23) strains. *B. bifidum*; ATCC, 11863 *L. plantarum*, FS2 and *P. pentosaceus*, D39 constitute the lactic acid bacteria (LAB) strains. Each bar is a mean of two independent replicates (*n* = 4) with its corresponding standard deviation. Bars with different letters (a–h) indicate significant differences (*p* < 0.05) according to Fisher’s LSD test.

**Figure 5 microorganisms-11-02942-f005:**
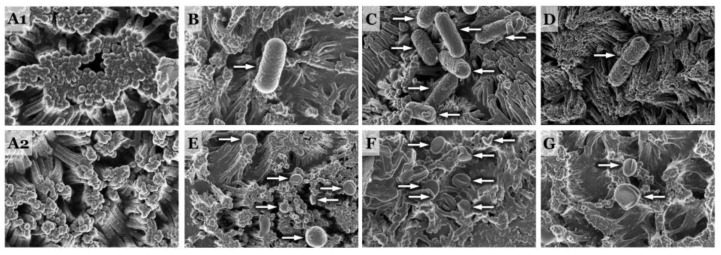
Scanning electron micrographs showing the adhesion of enteroaggregative *E. coli* and Lacid bacteria strains to differentiated Caco-2 (epithelial) monolayers. Plates (**A1**,**A2**) shows untreated differentiated Caco-2 cell monolayers whilst Plates (**B**–**D**) shows the adhesion of diarrhoeagenic enteroaggregative *E. coli* (D-EAEC) 3591-87 and K2, and then non-diarrhoeagenic enteroaggregative *E. coli* (ND-EAEC), N23, respectively. Plates (**E**–**G**) also shows Caco-2 cell monolayers infected with *B. bifidum*, ATCC 11863; *L. plantarum*, FS2, and *P. pentosaceus*, D39, respectively. The white arrows were either pointing at EAEC and LAB cells attached to the epithelial cells.

**Figure 6 microorganisms-11-02942-f006:**
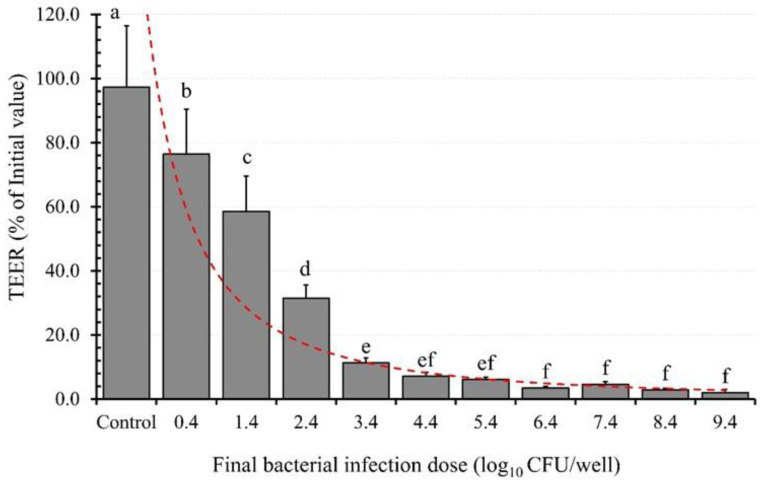
The effect of bacterial (enteroaggregative *E. coli*, EAEC, K2) infection dose on transepithelial electrical resistance (TEER) of polarized Caco-2 cell monolayers. Each bar is a mean of two independent replicates (*n* = 4) with its corresponding standard deviation. Bars with different letters (a–f) indicate significant differences (*p* < 0.05) according to Fisher’s LSD test.

**Figure 7 microorganisms-11-02942-f007:**
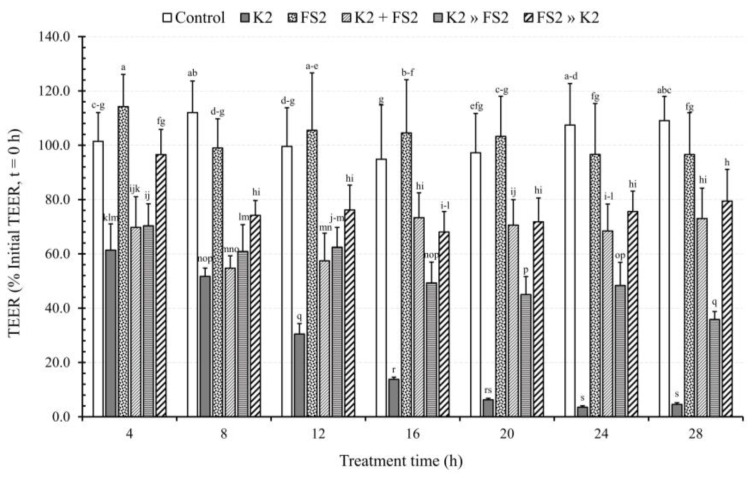
The effect of bacterial (enteroaggregative *E. coli*, EAEC, K2 and *L. plantarum*, FS2) infection mode and treatment time on transepithelial electrical resistance (TEER) of polarised Caco-2 cell monolayers. Each bar is a mean of two independent replicates (*n* = 4) with its corresponding standard deviation. Bars with different letters (a–s) indicate significant differences (*p* < 0.05) according to Fisher’s LSD test.

**Figure 8 microorganisms-11-02942-f008:**
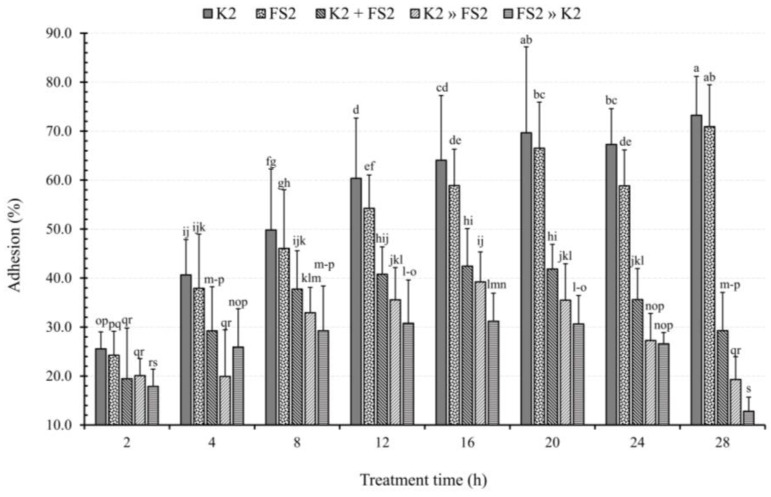
The effect of infection mode and treatment time on the adhesion of enteroaggregative *E. coli*, (EAEC), K2 and *L. plantarum*, FS2 to polarised Caco-2 cell monolayers. Each bar is a mean of two independent replicates (*n* = 4) with its corresponding standard deviation. Bars with different letters (a–s) indicate significant differences (*p* < 0.05) according to Fisher’s LSD test.

**Figure 9 microorganisms-11-02942-f009:**
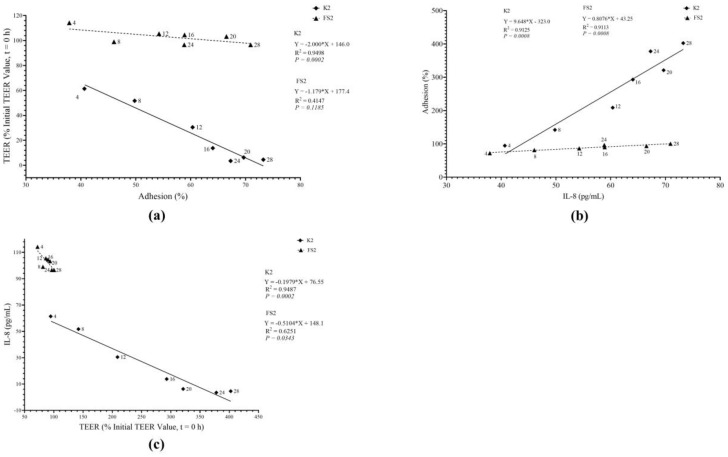
Relationship between bacterial (enteroaggregative *E. coli*, EAEC, K2 and *L. plantarum*, FS2) (**a**) adhesion and transepithelial electrical resistance (TEER); (**b**) adhesion and interleukin 8 (IL-8); (**c**) transepithelial electrical resistance (TEER) and interleukin 8 (IL-8) induction of Caco-2 monolayers. Each data point is a mean of two independent replicates (*n* = 4).

**Table 1 microorganisms-11-02942-t001:** Bacterial strains, sources, and culturing conditions for selected enteroaggregative *E. coli* (EAEC) and lactic acid bacteria (LAB) strains.

Bacteria Strain	Characteristic	Source
EAEC, 3591-87 ^a^	Clinical and diarrhoeagenic (positive reference strain)	NICD of NHLS ^c^
EAEC, K2 ^a^	Diarrhoeagenic	Unpasteurised fresh milk ^d^
EAEC, N23 ^a^	Non-Diarrhoeagenic	Unpasteurised fresh milk ^d^
*B. bifidum*, ATCC 11863 ^b^	Reference probiotic bacteria	ATTC Collections ^e^
*L. plantarum*, FS2 ^b^	Promising probiotic characteristics	Traditional fermented food (ogi) ^f^
*P. pentosaceus*, D39 ^b^	Promising probiotic characteristics	Traditional fermented food (ogi) ^f^

^a^ These strains were revived and cultured in tryptone soy broth, periodically plated on tryptone soy agar and on sorbitol McConkey agar for enumeration and incubated statically (37 °C, 18 h). ^b^ These LAB were revived and cultured in de Mann Rogosa and Sharpe (MRS) broth, plated on MRS agar for enumeration and incubated statically (37 °C, 18 h). ^c^ This EAEC strain was obtained from the National Institute for Communicable Diseases (NICD), a division of the National Health Laboratory Service (NHLS), Johannesburg, Republic of South Africa. ^d^ These strains were obtained as isolates from previous studies [30,31]. ^e^ Obtained from American Type Culture Collection (ATCC, USA). ^f^ Obtained as isolates from previous studies [32].

**Table 2 microorganisms-11-02942-t002:** Enteroaggregative *E. coli* (EAEC) and lactic acid bacteria (LAB) interaction template used for infecting Caco-2 cell monolayers for most experiments.

LAB Strains	EAEC Strains		
	3591-87	K2	N23
11863	3591-87 + 11863	K2 + 11863	N23 + 11863
FS2	3591-87 + FS2	K2 + FS2	N23 + FS2
D39	3591-87 + D39	K2 + D39	N23 + D39

Note: The EAEC strains include 3591-87, K2, K3, K16, and N23 and the LAB include *L. acidophilus*, ATCC 4356 and *Bifidobacterium bifidum*, ATCC 11863, *L. plantarum*, FS2, and *P. pentosaceus*, D39, respectively.

## Data Availability

The supporting data for the results of this study can be found at https://doi.org/10.25403/UPresearchdata.21746339.v1. Accessed on 15 November 2023.

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
