# Peer review of "Dynamic Interactions between Diarrhoeagenic Enteroaggregative *Escherichia coli* and Presumptive Probiotic Bacteria: Implications for Gastrointestinal Health"

_microorganisms, 2023, doi:10.3390/microorganisms11122942_

Round 1

Reviewer 1 Report

Comments and Suggestions for Authors

Author Response

Reviewer 1

I want to express my heartfelt appreciation to the first Reviewer (Reviewer 1) for his/her invaluable contributions that have greatly enriched this manuscript. I am deeply grateful for the constructive criticisms, insightful comments, and valuable suggestions provided, all of which have played a pivotal role in elevating the overall quality of the manuscript. The forthcoming rebuttals are crafted to specifically address the concerns raised during their meticulous review process.

Introduction

Comment / Query 1: “The introduction should provide a more detailed presentation of the pathogenicity, epidemiology, and clinical relevance of Enteroaggregative Escherichia coli (EAEC) bacteria, and the role of interleukin-8 (IL-8) in the context of EAEC infections.”

Rebuttal 1: The entire introduction section has been re-written detailing the issues raised in the comment.

Comment / Query 2: “Provide a comprehensive overview of EAEC as a pathogen. Discuss the diversity among EAEC strains, the key virulence factors associated with EAEC, and how these factors contribute to EAEC’s pathogenicity, and the global epidemiology of EAEC infections and their impact on public health.”

Rebuttal 2: The entire introduction section has been re-written detailing the issues raised in the comment.

Comment / Query 3: “Please, elaborate on the role of IL-8 as a proinflammatory cytokine in the context of gastrointestinal infections. Explain the significance of IL-8 in gastrointestinal health, how IL-8 is secreted by epithelial cells in response to infection and inflammation, how elevated IL-8 levels are often associated with inflammatory bowel diseases and various gastrointestinal infections, and why it is a key focus in this study.”

Rebuttal 3: The entire introduction section has been re-written, and the issues raised have been dealt with appropriately.

Comment / Query 4: “The citation format should be corrected in: The focus of this investigation is centred on the pivotal role of IL-8 in mediating hostpathogen interactions and the capacity of probiotics to mitigate its effects [14-17]. [18].”

Rebuttal 4: The entire introduction section has been re-written and the issues with the citation has been rectified.

Comment / Query 5: “and […] were isolated from unpasteurised fresh milk and a traditionally fermented West African cereal, Ogi, respectively [10]. [10-13].”

Rebuttal 5: The introduction section has undergone a complete revision, and the citation-related issues have been addressed and resolved.

Materials and Methods

Comment / Query 6: “The citation format should be corrected in: Obtained from previously isolated by Fayemi and Buys [11] Fayemi and Buys (2017).”

Rebuttal 6: The citation has been corrected in the specified section.

Comment / Query 7: “The citation is missing in:

Briefly, EAEC K2 cultures (18 h old) were standardized (1.5 x 109 CFU/mL) as previously

described (section 0).

[…] DMEM with the bacterial suspension followed by incubation (section 0).”

Rebuttal 7: The citation errors capturing “Section 0” have been rectified throughout the entire manuscript.

Comment / Query 8: “The challenged PCC-2CMNLs were incubated (37, 5% CO2) and assessed for their initial and final trans-epithelial resistances (4, 8, 12, 16, 20, 24 and 28 h) for the determination of TEER, whilst their corresponding supernatants were collected from the apical chambers and isolated (-20) for IL-8 assay as previously described [13].”

Rebuttal 8: The sentence herein has been changed to “The challenged PCC-2CMNLs were incubated (37, 5% CO2) and assessed for their initial and final TEERs (at 4, 8, 12, 16, 20, 24 and 28 h) whilst their corresponding supernatants were collected from the apical chambers and stored (-20 ℃) for IL-8 assay as previously described [34]”.

Comment / Query 9: “A few typo should be corrected: Which one is correct: Ω.cm2 or Ω cm2? With a dot or without a dot?”

Rebuttal 9: Thanks for this observation. Some authors use the dot whilst others use this expression without the dot but whatever the case may be, it should be used with consistency. The expression has been corrected throughout the manuscript without the dot.

Comment / Query 10: “[…] were then incubated at 37 ℃ with 5 % CO2 for 6 hours.”

Rebuttal 10: The expression, “were then incubated at 37 ℃ with 5 % CO2 for 6 hours.” has been corrected to read “were then incubated at 37°C in the presence of 5% CO2 for 6 hours.”

Results and Discussion

Comment / Query 11: “The results section of the manuscript presents a comprehensive analysis of the experiments conducted.”

Rebuttal 11: Thanks for your compliment.

Comment / Query 12: However, you mention “previous findings” or refer to prior studies several times. It would be helpful to briefly summarize or cite these studies to provide context for your results.

Rebuttal 12: The several sections affected were rectified by either briefly summarizing the studies or simply citing the related studies involved throughout the results and discussions section of the manuscript.

Comment / Query 13: “In the section 3.1 make sure that all bacteria names are italicized.”

Rebuttal 13: Thanks for that observation. All the affected bacterial names from section 3.1 and elsewhere have been properly italicized throughout the manuscript.

Comment / Query 14: “While the manuscript provides extensive data, it could enhance the discussion of the clinical implications and potential applications of the findings.”

Rebuttal 14: Thank you. A section discussing the clinical implications and potential applications of the findings has been written and positioned towards the end of the Results and Discussion section before the concluding section.

Comment / Query 15: “The limitations of the study, potential sources of bias, and areas for future research could be discussed in more detail.”

Rebuttal 15: Thank you. Limitations and biases of the current study, with an emphasis on areas for future research, have been addressed and positioned just before the concluding section of the manuscript.

Conclusion

Comment / Query 16: “This section is well-structured and provide a clear summary of the study's finding.”

Rebuttal 16: Thank you for your feedback.

Comment / Query 17: “References are properly formatted and cited in the manuscript.”

Rebuttal 17: Your feedback is highly appreciated.

Comment / Query 18: “The results section of the manuscript presents a comprehensive analysis of the experiments conducted. However, You mention "previous findings" or refer to prior studies several times. It would be helpful to briefly summarize or cite these studies to provide context for your results.”

Rebuttal 18: The concerns raised have been addressed substantially throughout the results and discussions section of the manuscript.

Reviewer 2 Report

Comments and Suggestions for Authors

This paper aimed to investigate the dynamic interactions between diarrhoeagenic enteroaggregative Escherichia coli and presumptive probiotic bacteria, which was of general interest to Microorganisms. The topic is interesting. However, the paper should be improved. I have listed some comments below:

1.       Page 4. It should be 75cm2, 2 should be superscripted. Please carefully review the issues related to superscripts throughout the entire manuscript.

2.       Section 2.3. The full name of DMEM has already been mentioned earlier, and its abbreviation can be used directly here.

3.       Please explain the reasons for choosing 6.0*107 of EAEC and 6.0*108 of LAB for the experiment. How were the dosages of both determined? Why are the dosages of the two not the same?

4.       Section 2.4. How is the time of 6 hours determined? Please cite relevant reference to support it.

5.       The authors should add a “Statistical analyzes” section to the last section of Materials and Methods.

6.       Figure 3. Please update the image resolution. The image is blurry, and some details cannot be seen clearly.

7.       References. There are too many references. Please reduce the references to about 60.

8.       The manuscript would benefit from review by an experienced (scientific) English writer.

Comments on the Quality of English Language

 Moderate editing of English language required.

Author Response

Reviewer 2

I wish to express my heartfelt appreciation to the second Reviewer (Reviewer 2) for his/her invaluable contributions that have greatly enriched this manuscript. I am deeply grateful for his/her constructive criticisms, insightful comments, and valuable input, all of which have played a pivotal role in elevating the overall quality of the manuscript. The ensuing responses aim to address the issues raised during the review process.

Comment / Query 1: “Page 4. It should be 75cm2, 2 should be superscripted. Please carefully review the issues related to superscripts throughout the entire manuscript.”

Rebuttal 1: Thank you. Issues relating to superscripts has been properly revised throughout the manuscript.

Comment / Query 2: “The full name of DMEM has already been mentioned earlier, and its abbreviation can be used directly here.”

Rebuttal 2: Thanks for your observation. The full name has been retracted leaving the DMEM alone.

Comment / Query 3: “Please explain the reasons for choosing 6.0*107 of EAEC and 6.0*108 of LAB for the experiment. How were the dosages of both determined? Why are the dosages of the two not the same?”

Rebuttal 3: The dosage for LAB was selected to be approximately 1 log10 higher than that for EAEC with the intention of preventing and treating EAEC infections. This approach aligns with the findings of previous researchers who advocated for a higher LAB dosage compared to EAEC. Alternatively, an equal dosage for both bacteria could be considered. The dosage was determined using McFarland densitometer (DEN-1 Model, Grant bio, Sia Biosan, Riga, Latvia) as detailed in “Section 2.4” of the manuscript.

Comment / Query 4: “Section 2.4. How is the time of 6 hours determined? Please cite relevant reference to support it.”

Rebuttal 4: The 6 hours was from the sentence “The challenged polarised Caco-2 cell monolayers (PCC-2CMLs) were then incubated at 37°C in the presence of 5% CO2 for 6 hours”. The 6 hours duration was determined by using a laboratory-based timer from the beginning to the end of the six (6) hours after which it was halted by removing the setup from the incubator to commence the other phase of the experiment.

Comment / Query 5: “The authors should add a “Statistical analyzes” section to the last section of Materials and Methods.”

Rebuttal 5: Thanks for your observation. Statistical section has been added to the manuscript appropriately.

Comment / Query 6: “Figure 3. Please update the image resolution. The image is blurry, and some details cannot be seen clearly.”

Rebuttal 6: The Figure 3 has been changed from the previous 300 to 768 dpi resolution.

Comment / Query 7: “References. There are too many references. Please reduce the references to about 60.”

Rebuttal 7: Thanks for the observation. The citations and references have been trimmed down to exactly 65.

Comment / Query 8: “The manuscript would benefit from review by an experienced (scientific) English writer.”

Rebuttal 8: Thanks for this observation. The entire manuscript has undergone review and feedback from four experienced scientific English writers. Their comments have been meticulously collated and integrated into the current version.

Round 2

Reviewer 2 Report

Comments and Suggestions for Authors

Authors addressed the previous questions and, by doing so, improved the manuscript.